# Natural History and Molecular Characteristics of Korean Patients with Mucopolysaccharidosis Type III

**DOI:** 10.3390/jpm12050665

**Published:** 2022-04-21

**Authors:** Min-Sun Kim, Aram Yang, Eu-seon Noh, Chiwoo Kim, Ga Young Bae, Han Hyuk Lim, Hyung-Doo Park, Sung Yoon Cho, Dong-Kyu Jin

**Affiliations:** 1Department of Pediatrics, Samsung Medical Centre, Sungkyunkwan University School of Medicine, Seoul 06351, Korea; min-sun.kim@samsung.com (M.-S.K.); aa.noh@samsung.com (E.-s.N.); gy22.bae@samsung.com (G.Y.B.); jindk.jin@samsung.com (D.-K.J.); 2Department of Pediatrics, Kangbuk Samsung Hospital, Sungkyunkwan University School of Medicine, Seoul 03181, Korea; dkfkal0718@hanmail.net; 3Department of Pediatrics, Soonchunhyang University Bucheon Hospital, Soonchunhyang University College of Medicine, Bucheon 14584, Korea; chngkak@nate.com; 4Department of Pediatrics, Chungnam National University College of Medicine, Daejeon 35015, Korea; damus@cnuh.co.kr; 5Department of Laboratory Medicine and Genetics, Samsung Medical Centre, Sungkyunkwan University School of Medicine, Seoul 06351, Korea; hyungdoo.park@samsung.com

**Keywords:** lysosomal storage disease, mucopolysaccharidosis III (MPS III), Sanfilippo syndrome, natural history

## Abstract

Background: Mucopolysaccharidosis type III (MPS III) is an autosomal recessive lysosomal storage disorder characterised by progressive neurocognitive deterioration. MPS III subtypes are clinically indistinguishable, with a wide range of symptoms and variable severity. The natural history of this disorder within an Asian population has not yet been extensively studied. This study investigated the natural history of Korean patients with MPS III. Methods: Thirty-four patients from 31 families diagnosed with MPS III from January 1997 to May 2020 in Samsung Medical Centre were enrolled. Clinical, molecular, and biochemical characteristics were retrospectively collected from the patients’ medical records and via interviews. Results: 18 patients had MPS IIIA, 14 had IIIB, and two had IIIC. Twenty (58.9%) patients were male. Mean age at symptom onset was 2.8 ± 0.8 years and at diagnosis was 6.3 ± 2.2 years. All patients with MPS IIIA and IIIB were classified into the rapidly progressing (RP) phenotype. The most common symptom at diagnosis was language retardation (88.2%), followed by motor retardation (76.5%), general retardation (64.7%), and hyperactivity (41.2%). Language retardation was more predominant in IIIA, and motor retardation was more predominant in IIIB. The mean age of the 13 deceased patients at the time of the study was 14.4 ± 4.1 years. The age at diagnosis and lag time were significantly older and longer in the non-survivor group compared with the survivor group (*p* = 0.029 and 0.045, respectively). Genetic analysis was performed in 24 patients with MPS III and identified seven novel variants and three hot spots. Conclusion: This study is the first to analyse the genetic and clinical characteristics of MPS III patients in Korea. Better understanding of the natural history of MPS III might allow early diagnosis and timely management of the disease and evaluation of treatment outcomes in future clinical trials for MPS III.

## 1. Introduction

Mucopolysaccharidosis type III (MPS III or Sanfilippo syndrome) is an autosomal recessive lysosomal storage disorder characterised by progressive neurocognitive deterioration [1]. MPS III is caused by a deficiency of one of four enzymes involved in heparan sulphate (HS) degradation: heparin N-sulfatase (*SGSH*, MPS IIIA, MIM #252900), α-N-acetylglucosaminidase (*NAGLU*, MPS IIIB, MIM #252920), acetyl-CoA α-glucosaminide acetyltransferase (*HGSNAT*, MPS IIIC, MIM #252930), or N-acetylglucosamine-6-sulfatase (*GNS*, MPS IIID, MIM #252940) [2]. MPS IIIA and IIIB are the common subtypes [3,4]; MPS IIIC is less common, and MPS IIID is the rarest [5]. The estimated incidence of MPS III is 1 in 70,000 live births [6], and it varies according to subtype and geographical region [7]. In northern Europe (including France, Germany, the Netherlands, Sweden, and the UK), MPS IIIA is the most prevalent subtype; however, in southern countries (including Greece and Portugal), MPS IIIB predominates [8]. In Asia, MPS IIIB is dominant, according to studies in Taiwan [9] and China [10].

Patients with MPS III develop normally during the first year of life. Neurodegenerative symptoms occur due to accumulation of HS within the central nervous system (CNS), resulting in cognitive decline, motor dysfunction, and behavioural abnormalities [11]. Somatic effects such as facial coarseness, hearing loss, hepatosplenomegaly, and dysostosis multiplex are less pronounced than in other forms of MPS [12,13]. MPS III subtypes are clinically indistinguishable, with a wide range of symptoms and variable severity. MPS III can be further divided into rapidly progressing (RP) and slowly progressing (SP) phenotypes.

There is currently no effective treatment for MPS III, and only conservative management is possible. Disease-specific treatments for MPS IIIA and IIIB are being studied, including intrathecally delivered enzyme replacement therapy (ERT), substrate reduction therapy (SRT), hematopoietic stem cell transplantation, and gene therapy [7,14,15,16]. A few studies on the natural history of MPS III have been reported in Caucasian populations [8,12,17]. However, the natural history of this disorder within an Asian population has not yet been extensively studied. In this study, we investigate the natural history of MPS III in Korean patients and the clinical differences between IIIA and IIIB of the RP phenotype and between surviving and deceased groups.

## 2. Materials and Methods

Thirty-four patients from 31 families with MPS III diagnosed from January 1997 to May 2020 in Samsung Medical Centre were enrolled in this study. All patients were diagnosed with MPS IIIA, IIIB, or IIIC by enzymatic activities in leukocytes and/or fibroblasts. In most cases, identification of pathogenic mutations was performed. Clinical, molecular, and biochemical characteristics were retrospectively collected from patients’ medical records and via interviews. Multiple answers were permitted during the interview. One investigator conducted the database analysis and the face-to-face or telephone interviews. Demographic information including gender, age at diagnosis, age at the first seizure, age at death, cause of death, molecular and laboratory results, ambulatory status at the time of the latest medical records, physical examinations, and surgical interventions were collected. Any available results of the following investigations were also collected: brain magnetic resonance imaging (MRI), echocardiography, abdominal ultrasonography, and other investigations relevant to the disease. The Four-Point Scoring System (FPSS) [18] was administered as an MPS III-specific parent-completed disability questionnaire that rates motor, expressive language, and cognitive function on a 3-point scale (normal = 3, beginning regression = 2, severe regression = 1, and lost skill = 0). The interviews and scoring assessments were all conducted by the same interviewer. The phenotype of MPS IIIA and IIIB is determined by age at diagnosis before (RP) or after (SP) age 6 years, which is a reliable indicator of disease severity [1,8] and/or known genotype severity. The RP phenotype with MPS IIIC includes loss of speech and motor ability, generally occurring within 1-5 years of each other [19]; the SP phenotype progresses more slowly, as one could anticipate by its designation. Inability to associate three words at 15 months was noted as language retardation according to the Centers for Disease Control and Prevention (CDC) language milestones [20]. Motor retardation was defined as inability to walk alone at age 15.3 months or older (95th percentile) according to gross motor milestones of the World Health Organization (WHO) [21].

### 2.1. MPS III Subtype Classification: Enzyme Assays and Genetic Tests

Enzyme activities were measured using fluorogenic substrates. The reference range of each enzymatic activity is as follows: Heparan N-sulfatase (IIIA): 8.9–64 pmol/min/mg protein (skin fibroblast), 4.1–12 nmol/17h/mg protein (leukocyte); α-N-acetylglucosaminidase (IIIB): 0.90–1.51 nmol/h/mg protein (leukocyte), 22.3–60.9 nmol/h/mL protein (plasma); acetyl-CoA α-glucosaminide acetyltransferase (IIIC), 8.6–32 nmol/17h/mg protein (leukocyte). After confirmation by enzyme assays, a Sanger sequencing test was performed to search for gene mutations related to the target subtype of MPS III.

### 2.2. Statistical Analysis

To compare continuous variables, a parametric independent two-sample t-test was conducted. The Mann–Whitney U-test was used for nonparametric variables based on the normality assumptions from the Shapiro–Wilk test. The chi-square test or Fisher’s exact test was used to compare categorical variables. The Kaplan–Meier survival curve was used for the survival rate of patients with MPS III and was compared between groups with the use of the log-rank test. Analysis was performed using R Statistical Package (version 3.6.2; Institute for Statistics and Mathematics, Vienna, Austria). A *p*-value < 0.05 was considered statistically significant.

### 2.3. Ethics Statement

The study was approved by the Institutional Review Board (IRB) of Samsung Medical Centre, Sungkyunkwan University School of Medicine, Seoul, Korea (IRB no. 2020-10-164-003). Informed consent was obtained from the patients and/or their parents.

## 3. Results

Data were collected from 34 Korean patients (58.9% males and 41.1% females) with MPS III from 31 families (Table 1). MPS IIIA, IIIB, and IIIC were present in 52.9% (*n* = 18), 41.2% (*n* = 14), and 5.9% (*n* = 2) of MPS patients, respectively. Enzyme activities in all patients were very low or undetectable, at less than 10% of the lower limit of the reference value. The mean age at symptom onset was 2.8 ± 0.8 years, and the mean age at diagnosis was 6.3 ± 2.2 years. The most common symptom at diagnosis was neurodevelopmental retardation. The most common initial symptom was language retardation (88.2%), followed by motor retardation (76.5%), both language and motor retardation (64.7%), and behavioural abnormalities (hyperactivity, aggression, and unawareness of dangerous situations) (41.2%) (Table 1 and Table 2). In language milestones, only four children (11.8%) acquired the capacity to associate three words by age 15 months. The performance was not significantly different according to MPS III subtype, except patients with IIIB who started walking later than those with IIIA (*p* = 0.002). All patients with MPS IIIA or IIIB were diagnosed before the age of 6 years and/or associated with the severe genotype; therefore, all of them were classified into the RP phenotype. The mean diagnosis lag time (time interval between symptom onset and diagnosis) was 3.6 ± 2.5 years (Table 1). 

### 3.1. Clinical Presentation

There were no differences in clinical characteristics between patients with MPS IIIA and IIIB except for age at first walking. The median age at first walking without assistance was 1.25 years (IQR, 1–1.37) in IIIA and 1.55 years (IQR, 1.42–1.6) in IIIB (Table 1). Language retardation in IIIA and motor retardation in IIIB were the most common initial symptoms. Patients with IIIB started walking later than patients with IIIA (*p* = 0.01) (Table 2). In patients with MPS III, the most common accompanying symptoms after diagnosis were behavioural abnormalities and impairment of speech (Table 2). Frequent falls due to clumsy walking occurred in 91.2% of patients. The data included one case of a forehead laceration and two cases of brain haemorrhages due to falls. Hearing impairment was reported in 17 patients (50%), of whom seven used hearing aids. Thirty patients (88.2%) showed coarse facial features, hirsutism (85.3%), or Mongolian spots (64.7%). More than half (55.9 %) of the patients with MPS III were tested for heart disease. Mitral regurgitation, tricuspid regurgitation, and mitral valve thickening were detected in 78.9%, 84.2%, and 68.4% of respective patients, respectively, at the time of data collection. Among 21 patients who underwent abdominal ultrasonography, organomegaly was observed in 17 (80.9%) (Table 1), and hernias were found in eight (23.5%). A box plot figure showing the clinical features of all 34 patients is presented as Appendix A. Twenty-four patients (70.6%) underwent at least one surgical procedure. The most prevalent surgical interventions were gastrostomy (52%), tracheostomy (17%), herniorrhaphy (11%), tonsillectomy (11%), adenoidectomy (11%), and craniostomy (11%), followed by ear tube insertion (5%), scrotal hydrocelectomy (5%), and frenulotomy of tongue (5%) (Figure 1).

In this study, two patients with MPS IIIC were siblings. The younger sibling [3] was allergic to milk formula and food, so she consumed breast milk exclusively for over one year and was diagnosed with vitamin D-deficient rickets. The patient showed mild coarse facial features, hepatomegaly, delayed motor development, and dysplastic vertebrae at 2 years. However, cognitive function and speech development were normal at that time. The older sibling showed delayed speech development and normal motor development. Neither sibling showed hyperactivity. The urinary GAG levels of the older and younger siblings were 502.5 and 984.4 mg GAG/g creatinine (reference range: <175 mg GAG/g creatinine), respectively, and elevated GAG was identified as heparan sulphate by thin-layer chromatography. Additionally, the *HGSNAT* activities of the siblings were 0.6 and 0.7 nmol/17 h/mg protein (reference range, 8.6–32 nmol/17 h/mg protein), respectively. The younger sibling was diagnosed with MPS IIIC at 2.3 years, while the older was diagnosed only at 5.1 years. The siblings showed the SP phenotype, with loss of speech and walking ability occurring after age 10 years. 

### 3.2. Neurodegenerative Symptoms

All patients of MPS IIIA and IIIB were classified into the RP phenotype, showing rapid cognitive decline at a mean age of 4.4 ± 2.7 years. The clinical course of MPS IIIA and IIIB was tracked using the FPSS (Figure 2). The onset of speech regression (score of 2) was observed at a median age of 3.9 years (range, 2.1–5 years), whereas motor and cognitive functions began to regress at a median age of 5.4 years (range, 3–13 years) and 4.4 years (range, 1.5–11 years), respectively. Severe regression (score of 1) in speech, motor, and cognitive function was found at a median age of 7.3 years (range, 2–10 years), 10.4 years (range, 7–15 years), and 8.2 years (range, 5–13 years), respectively. Approximately 78.8 % of patients presented with epilepsy at a mean age of 10.0 ± 2.9 years. Convulsions were usually controlled with one or two antiepileptic drugs (AEDs). Age at loss of speech in MPS III subtypes is shown in Appendix A. Clumsiness in walking was observed starting at a median age of 10 years (range, 4.6–15 years). Data on the loss of walking ability are shown in Appendix A. Dysphagia began at a median age of 12 years (range, 5.4–16 years). At a mean age of 15.1 ± 1.9 years, the patients were in a fully bedridden state. Gastrostomy (18/34, 52.9%) was performed at a mean age of 17.2 ± 1.9 years, and tracheostomy was performed at a mean age of 17.8 ± 3.7 years. In siblings with MPS IIIC, the onset of speech regression (score of 2) was observed at a mean age of 10.5 ± 0.5 years. Motor functions and cognitive functions began to regress (score of 2) at the age of 11.5 ± 0.75 years and 11.0 ± 0.5 years, respectively.

### 3.3. Survival Analysis by Subtype

The mean follow-up period for MPS IIIA and IIIB was 12.6 ± 6.1 year. Of the 18 patients with MPS IIIA, six (33.3%) died; of the 14 patients with MPS IIIB, seven (50%) died. The mean age at death was 14.4 ± 4.1 years; these patients died of pneumonia (*n* = 8, 11.3–23.9 years), epilepsy (*n* = 1, 10.1 years), or of an unknown cause (*n* = 4, 11.4–21.4 years). When the survival rates between the MPS IIIA and IIIB groups were compared, there was no significant difference (*p* = 0.89) (Figure 3). Table 3 presents the comparison between the 13 patients who died and 21 survivors at the time of this study. Five patients (38.5%) who died were female, but the sex difference was not statistically significant. The age at symptom onset was similar between the groups, but the age at diagnosis was significantly older in the non-survival group (7.3 ± 1.8 years compared with 5.6 ± 2.3 years in the survivor group; *p* = 0.029). This indicates a significant increase in lag time to diagnosis in the non-survivor group (4.8 ± 3.0 versus 2.8 ± 1.8 years, *p* = 0.045). History of AED medication was significantly more frequent in the non-survivor group (100% versus 61.9%, *p* = 0.027).

### 3.4. Genetic Analysis

Genetic analysis was performed in 24 patients with MPS III. Most of the mutation types were missense mutations: 92.3% in *SGSH* mutations and 88.9% in *NAGLU* mutations (Figure 4A). In addition, the *NAGLU* mutations were mostly located in exon 6 (83.3%) and the *SGSH* mutations were mostly located in exon 6 (50%) and exon 8 (50%) (Figure 4B,C). In patients with MPS IIIA, seven known mutation alleles were found: c.703G > A (p.Asp235Asn), c.812C > T (p.Thr271Met), c.1040C > T (p.Ser347Phe), c.544C > T (p.Arg192Cys), c.1129C > T (p.Arg377Cys), c.449G > A (p.Arg150Gln), and c.823G > A (p.Gly275Arg). Four alleles were novel variants of likely pathogenic nature: c.703G > C (p.Asp235His), c.69delG (p.Asn24Thrfs*240), c.1094A > G (p.Gln365Arg), and c.228C > G (p.Ser76Arg). We found six mutant alleles in nine patients with MPS IIIB including the three known mutations c.1444C > T (p.Arg482Trp), c.1694G > C (p.Arg565Pro), and c.607C > T (p.Arg203*). Three were novel variants: c.1976C > T (p.Ala659Val), c.200T > C (p.Leu67Pro), and c.775C > T (p.Gln259*). The siblings with MPS IIIC were found to be compound heterozygous for two known *HGSNAT* mutations: c.234+1G > A (IVS2+1G > A) and c.1150C > T (p.Arg384*). The location and allele frequency of all mutations are shown in Figure 4. In our study, c.1444C > T (p.Arg482Trp) was a hot spot for MPS IIIB (61.1%), while c.1040C > T (p.Ser347Phe) was found in 26.9% and c.703G > A (p.Asp235Asn) in 23.1% of MPS IIIA (Figure 5A,B).

## 4. Discussion

This is the first study to describe the natural history and genetic features of MPS III in Korea. In this study, MPS IIIA was the most common type (52.9%), followed by type IIIB. IIIB is the most common in Taiwan (82.1%) [9] and in China (55.9%) [10], which is different from Korea. In the Asian population, there have been only two studies regarding the natural history of MPS III in China [10] and Taiwan [9]. These studies determined that speech delay and intellectual disability were the most prevalent clinical manifestations, followed by hyperactivity, similar to our cohort study. In the literature, no clinical differences in development were found between MPS IIIA and IIIB. In this study, we additionally evaluated motor development using age at first walking without assistance and noted a difference between the MPS IIIA and IIIB groups.

The first stage of MPS III, delay in speech and cognitive development, usually occurs between the ages of 1 and 3 years for RP phenotypes and around the age of 4 years for SP phenotypes [17]. In RP phenotypes, the first stage is immediately followed by a period of progressive neurocognitive decline, emergence of behavioural difficulties, and sleep disturbances. Patients with MPS IIIA and IIIB usually die within the first 2 decades of life. MPS IIIC has an onset in early childhood, with death generally occurring before the third decade [19]. Of the four subtypes, the RP phenotype is most frequently reported in MPS IIIA [13]. MPS IIIB is quite similar to IIIA, but more attenuated patients with this phenotype have been reported [22]. In our cohort, all patients presented with the RP phenotype, and there were no significant differences in survival rates between patients with IIIA and those with IIIB (Figure 3). However, language retardation was more predominant in IIIA, and motor retardation was more predominant in IIIB (Table 2). There was a significant difference in motor retardation (*p* = 0.01) (Table 2). MPS IIIA patients manifested epilepsy before age 10, whereas this occurred at a later age for patients with MPS IIIB. Epilepsy was usually well controlled with one or two AEDs in most patients. The younger sibling with MPS IIIC was diagnosed at the age of 2.3 years, which was much earlier than most diagnoses, even considering the wide range in the average age at diagnosis with IIIC (4.5 to 12 years) [8,23,24,25] and the known mean age at symptom and sign onset (3.5 years) [6].

A total of 13 patients died during the study period, and there was a diagnostic delay of 7.3 ± 1.8 years in the mean age at diagnosis. The mean age at death for our patients with IIIA and IIIB was 14.4 ± 4.1 years, which is in line with previous studies of the natural history of MPS III [12,17,18,26,27] (Table 4). When the group of patients who had died at the time of the study was compared with the survivor group (Table 3), the mean age at symptom onset was similar between the two groups. Furthermore, the mean age at diagnosis for the survivor group was significantly younger (5.6 ± 2.3 years vs. 7.3 ± 1.8 years, *p* = 0.029), and the lag time to diagnosis in the survivor group was also significantly shorter than that of the non-survivor group (2.8 ± 1.8 years vs. 4.8 ± 3.0 years, *p* = 0.045). The delay in diagnosis of MPS III might be due to poor awareness caused by the rarity of the disease, the absence of minor somatic manifestations compared with the other types of MPS, and/or the main symptoms of neurologic regression (e.g., speech delay and behavioural problems) being similar to those of other neurologic disorders [28] such as idiopathic developmental delay, attention deficit hyperactivity disorder, and autism spectrum disorder [29,30,31]. In our cohort, 18 patients (100%) with IIIA and 11 patients (78.6%) with IIIB showed language retardation as an initial symptom. In addition, 11 (61.1%) patients with IIIA and 14 (100%) patients with IIIB showed motor retardation as an initial symptom. The median age at first walking was 1.25 years (IQR, 1–1.37) in IIIA and 1.55 years (IQR, 1.42–1.6) in IIIB (Table 1). Parents might tend to be more sensitive to language retardation than motor retardation, and a longer median diagnosis lag time was observed for MPS IIIB in our cohort (Table 1).

In this study, all patients with MPS IIIA and IIIB were classified into the RP type, and no mutations related to SP type were found. According to the Leiden open variation database of *SGSH* [32] and *NAGLU* [33], the total public variants reported for the *SGSH* gene are 217, and 112 were reported for *NAGLU* gene (last updated 17 September 2021), of which the majority are missense mutations. All novel variants in this study were predicted to be pathogenic by SIFT, Mutation Taster, and Polyphen-2 prediction software. The novel variants comprise 26.9% of the alleles for the *SGSH* gene and 16.7% of the alleles for the *NAGLU* gene (Table 5).

Specific mutations in *SGSH* associated with the RP type are related to distinct geographical locations. p.Arg245His, p.Arg74Cys, 1091delCys, and p.Ser66Trp were the most frequent mutations in the Dutch (56.7%) [34], Polish (56%) [35], Spanish (45.5%) [36], and Italian (33%) [37] populations, respectively. In this study, p.Ser347Phe [38] and p.Asp235Asn [39,40] mutations were common (26.9% and 23.1% of alleles, respectively). The Asp235Asn mutation was reported to belong to the RP phenotype [40], whereas the Ser347Phe mutation has not been related to any phenotype [37,40]. p.Arg377Cys, p.Asn24Thrfs*240, p.Gln365Arg, p.Arg150Gln, and p.Asp235Asn [40] in *SGSH* (Figure 5) belong to the RP type of MPS IIIA.

**Table 5 jpm-12-00665-t005:** Mutations found in the *SGSH, NAGLU*, and *HGSNAT* genes from the 24 MPS III index cases.

Gene	Reference Sequences	DNA Nucleotide Change	Predicted Protein Change	Common [Reference]	Patient Number
*SGSH*	NM_000199.5 NP_000190.1	c.1129C > T	p.Arg377Cys	Di Natale et al. [37]	9A
		c.1094A > G	p.Gln365Arg	Novel	9A
		c.1040C > T	p.Ser347Phe	Miyazaki et al. [38]	1A, 2A 3A, 4A, 7A, 10A ^£^
		c.823G > A	p.Gly275Arg	Heron et al. [8]	15A
		c.812C > T	p.Thr271Met	Heron et al. [8]	5A
		c.703G > A	p.Asp235Asn	Beesley et al. [41]	1A, 4A, 5A, 12A, 14A, 17A
		c.703G > C	p.Asp235His	Novel	3A, 7A, 12A
		c.544C > T	p.Arg192Cys	Di Natale et al. [37]	2A
		c.449G > A	p.Arg150Gln	Bunge et al. [35]	15A, 17A
		c.228C > G	p.Ser76Arg	Novel	14A
		c.69delG	p.Asn24Thrfs*240	Novel	8A^£^
*NAGLU*	NM_000263.4 NP_000254.2	c.1976C > T	p.Ala659Val	Novel	1B
		c.1694G > C	p.Arg565Pro	Weber et al. [42]	12B, 14B ^£^
		c.1444C > T	p.Arg482Trp	Bunge et al. [43]	1B, 2B ^£^, 5B ^£^, 6B, 8B, 11B ^£^, 13B ^£^
		c.775C > T	p.Gln259*	Novel	12B
		c.607C > T	p.Arg203*	Schmidtchen et al. [44]	8B
		c.200T > C	p.Leu67Pro	Novel	6B
*HGSNAT*	NM_152419.3 NP_689632.2	c.234+1G > A	IVS2+1G > A	Canals et al. [45]	1C, 2C
		c.1150C > T	p.Arg384*	Ruijter et al. [23]	1C, 2C

^£^ homozygous mutation.

p.Arg565Trp [46], p.Arg565Gln, and p.Arg565Pro have been reported as mutational hotspots in *NAGLU*. In this cohort, p.Arg565Pro accounts for 16.7% of all alleles detected, while p.Arg482Trp [43,47] accounts for 61.1% and is the hotspot [48]. p.Arg565Pro [43], p.Arg482Trp [42,49], p.Gln259*, and p.Arg203* in *NAGLU* were associated with the RP phenotype. Both p.Arg565Pro and p.Arg482Trp are located in exon 6, and all mutations found in exon 6 account for 83.3% of the total alleles. Therefore, exon 6 is assumed to be an important site in Korean patients with MPS IIIB. The IVS2+1G > A and p.Arg384* mutations in *HGSNAT* have been frequently reported globally [3,23,50,51].

Currently, several clinical trials are investigating different disease-modifying treatment options for MPS IIIA and IIIB. As a disease modifying treatment of MPS IIIA is imminent, early assessment of the phenotype (RP or SP), i.e., identification before progression of clinical signs and symptoms, is crucial both for the inclusion of patients in trials and for the assessment of treatment effects based on the expected natural history of the disease. De Ruijter et al. [52] showed that newborn screening for MPS IIIA was feasible by measuring HS concentrations or lysosomal protein concentrations in dried blood spots. However, the current lack of consensus regarding when to begin treatment is an ongoing problem, and anxiety is a common issue in parents due to positive screenings for pseudodeficiencies (false positive) or for variants of unknown significance (VUS).

This study has several limitations. First, this study was retrospective. Nevertheless, the amount of missing data was small, and the data were verified via telephone interviews and assessment of medical records. Second, the number of patients included in this study was small. However, this study was performed in a single centre, and our institution has managed most of the MPS population in Korea; therefore, our data reliably represent the general situation in Korea. All patients were managed by the same protocol, allowing consistent data analysis.

## 5. Conclusions

This is the first study that analysed the genetic and clinical characteristics of MPS III patients in Korea. MPS III should be considered when a patient shows neurodegenerative symptoms such as language retardation, motor retardation, or hyperactivity. MPS IIIA is the most predominant in Korea. There were no differences in clinical characteristics between patients with MPS IIIA and IIIB except for age at first walking. Early diagnosis is important to increase the survival rate for this population, and new therapeutic strategies will significantly change the course of the disease. Our findings can be used to develop quality of care strategies and provide guidance for clinical trial endpoint evaluations. 

## Figures and Tables

**Figure 1 jpm-12-00665-f001:**
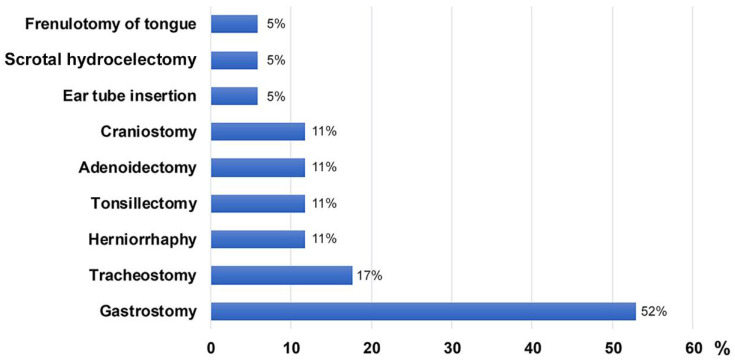
Types of surgical interventions for 34 patients with MPS III in Korea.

**Figure 2 jpm-12-00665-f002:**
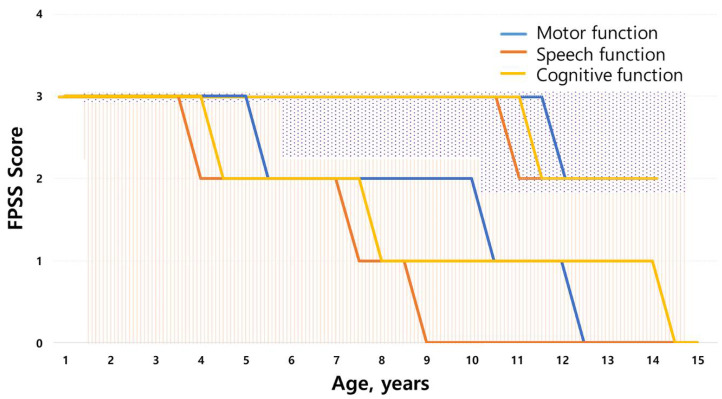
Scoring evaluation of the natural course of MPS III using the Four-Point Scoring System (FPSS). Regression of abilities as assessed by the FPSS (0–3) divided into average age score for motor function, speech, and cognitive function of the MPS III study population (*N* = 34). The graph line on the dotted background is the score for patients with MPS IIIC, and the graph line on the striped background is the score for patients IIIA and IIIB.

**Figure 3 jpm-12-00665-f003:**
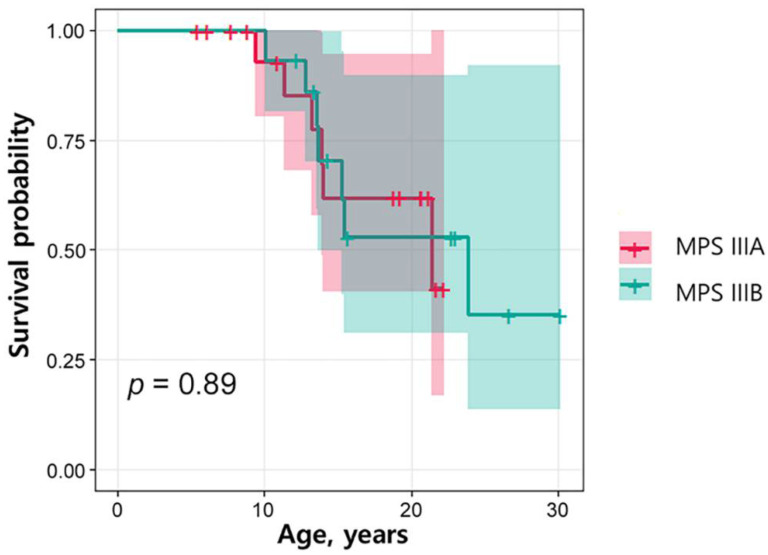
Kaplan–Meier survival estimates according to MPS IIIA and IIIB. The Kaplan–Meier analysis curve of the survival rate of patients with MPS IIIA and IIIB was compared between groups with the use of the log-rank test. There was no significant difference (*p* = 0.89).

**Figure 4 jpm-12-00665-f004:**
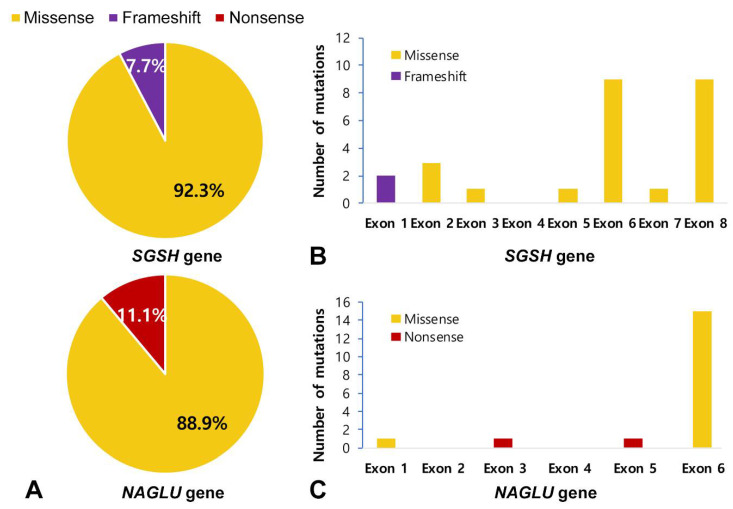
Mutation type profile and Exon distribution of MPS III. (**A**) Mutation type profile in patients with MPS IIIA and IIIB. (**B**,**C**) Exon distribution of the *SGSH* and *NAGLU* mutations in this study. Most were missense mutations (92.3% in *SGSH* mutations, 88.9% in *NAGLU* mutations).

**Figure 5 jpm-12-00665-f005:**
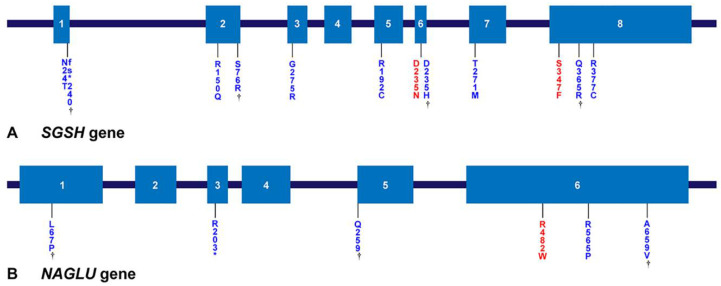
Locations of gene mutations of MPS IIIA and IIIB. (**A**,**B**) Locations of mutations in the *SGSH* gene and the *NAGLU* gene. Exons are represented by boxes indicated by Arabic numerals. The number of amino acids encoded by the individual exons and the approximate length of each intron are shown. The mutations found in this study are shown below the gene; mutations marked in red are the hotspot mutations most often found in this study, and genes marked with a dagger are novel mutations found in this study.

**Table 1 jpm-12-00665-t001:** Clinical characteristics of patients with MPS III.

Characteristic	Total (*N* = 34)	IIIA (*N* = 18)	IIIB (*N* = 14)	*p*-Value	IIIC (*N* = 2)
Sex				>0.99	
Male	20 (58.9%)	11 (61.1%)	8 (57.1%)		1 (50%)
Female	14 (41.1%)	7 (38.9%)	6 (42.9%)		1 (50%)
Follow-up period (mean years ± SD)	12.6 ± 6.1	12.3 ± 5.8	13.6 ± 6.5	0.58	11.2 ± 0.6
Age at symptom onset (mean years ± SD)	2.8 ± 0.8	3.1 ± 0.9	2.6 ± 0.7	0.10	2.0 ± 0.2
Age at diagnosis (mean years ± SD)	6.3 ± 2.2	6.3 ± 2.3	6.5 ± 2.0	0.77	3.7 ± 1.4
Diagnosis lag time (mean years ± SD)	3.6 ± 2.5	3.5 ± 2.9	3.9 ± 1.9	0.29	1.7 ± 1.2
Cardiac evaluation by TTE (positive finding/test available, %)					NA
MR	15/19 (78.9%)	10/14 (71.4%)	5/5 (100.0%)	0.53	
TR	16/19 (84.2%)	11/14 (78.6%)	5/5 (100.0%)	0.53	
MV thickening	13/19 (68.4%)	9/14 (64.3%)	4/5 (80.0%)	>0.99	
Abdominal US (positive finding / test available, %)					
Hepatosplenomegaly	17/21 (80.9%)	9/12 (75%)	6/7 (85.7%)	>0.99	2/2 (100%)
SMA syndrome	6/21 (28.6%)	4/12 (33.3%)	2/7 (28.6%)	0.51	0
Clinical findings					
Age at first walking without assistance, median years (IQR Q1,Q3)	1.3 (1.23, 1.6)	1.25 (1, 1.37)	1.55 (1.42, 1.6)	0.002 *	1.3, 1.0
Age of first seizure (mean years ± SD)	10.0 ± 2.9	9.8 ± 2.8	10.3 ± 3.2	0.739	NA
History of AEDs medication	26 (76.5%)	15 (83.3%)	11 (78.6%)	>0.99	0
Gastrostomy	18 (52.9%)	9 (50%)	9 (64.3%)	0.65	0
Age at gastrostomy (mean years ± SD)	17.2 ± 1.9	16.8 ± 2.2	17.6 ± 1.8	0.421	NA
Tracheostomy	6 (17.6%)	4 (22.2%)	2 (14.3%)	0.67	0
Age at tracheostomy (mean years ± SD)	17.8 ± 3.7	16.3 ± 1.3	21 ± 5.7	0.24	NA
Deafness	17 (50%)	11 (64.7%)	4 (30.8%)	0.14	2 (100%)
Bed-ridden status	21 (61.8%)	12 (57.1%)	9 (42.9%)	>0.99	NA
Age at onset of bed-ridden state (mean years ± SD)	15.1 ± 1.9	15.3 ± 1.6	14.9 ± 2.4	0.667	NA

Abbreviations: SD = standard deviation; IQR = interquartile range; TTE = transthoracic echocardiography; NA = not available; MR = mitral regurgitation; TR = tricuspid regurgitation; MV = mitral valve; US = ultrasonography; SMA = superior mesenteric artery syndrome; AEDs = antiepileptic drugs; * *p* < 0.05.

**Table 2 jpm-12-00665-t002:** First symptoms observed and associated symptoms after diagnosis in patients with MPS III (*N* = 34).

First Symptom ^a^	Total, *N* (%)	IIIA, *N* (%)	IIIB, *N* (%)		*p*-Value	IIIC, *N* (%)
Language retardation	30 (88.2)	18 (100)	11 (78.6)		0.073	1 (50)
Motor retardation	26 (76.5)	11 (61.1)	14 (100)		0.010 *	1 (50)
Language and motor retardation	22 (64.7)	11 (61.1)	11 (78.6)		0.721	0 (0)
Behavioural abnormalities	14 (41.2)	8 (44.4)	6 (42.9)		>0.99	0 (0)
Coarse facial features	6 (17.6)	3 (16.7)	2 (14.3)		>0.99	1 (50)
Skeletal abnormalities	4 (11.7)	2 (16.7)	1 (7.1)		>0.99	1 (50)
Hearing loss	4 (11.7)	1 (5.6)	2 (14.3)		0.568	1 (50)
**Associated Symptom ^a^ after Diagnosis**	***N* (%)**			**Median Age, Year (Range)**		
Impairment of speech	33 (97.1)			3.9 (1.5–5.0)		
Behavioural abnormalities	32 (94.2)			4.0 (0.5–7.0)		
Clumsy walking	31 (91.2)			10.0 (4.6–15.0)		
Macrocephaly and coarse face	30 (88.2)			3.0 (1.0–4.0)		
Sleep disorder	30 (88.2)			3.7 (0.5–6.6)		
Hirsutism	29 (85.3)					
Mongolian spots	22 (64.7)					
Hearing loss	17 (50.0)			5.0 (0–8.5)		
Recurrent otitis	13 (38.2)			2.5 (1.0–4.0)		
Recurrent diarrhoea	8 (23.5)			3.5 (0.5–5.5)		
Hernia	8 (23.5)			0.6 (0–1.2)		

^a^ Multiple answers were permitted; * *p* < 0.05.

**Table 3 jpm-12-00665-t003:** Comparison of survivor and non-survivor groups in patients with MPS III.

Characteristic	Survivor (*N* = 21)	Non-Survivor (*N* = 13)	*p*-Value
Sex			>0.99
Male	12(57.2%)	8 (61.5%)	
Female	9 (42.8%)	5 (38.5%)	
Type			0.68
IIIA	12 (57.2%)	6 (46.2%)	
IIIB	7 (33.3%)	7 (53.8%)	
IIIC	2 (9.5%)	0 (0%)	
Age at death (mean years ± SD)		14.4 ± 4.1	
Age at diagnosis (mean years ± SD)	5.6 ± 2.3	7.3 ± 1.8	0.029 *
Age at symptom onset (mean years ± SD)	2.8 ± 0.8	2.9 ± 0.9	0.69
Lag time to diagnosis (mean years ± SD)	2.8 ± 1.8	4.8 ± 3.0	0.045 *
Cardiac evaluation by TTE	11 (100%)	8 (100%)	
MR	8 (72.7%)	7 (87.5%)	0.60
TR	9 (81.8%)	7 (87.5%)	>0.99
MV thickening	6 (54.6%)	7 (87.5%)	0.18
Abdominal US	14 (100%)	7 (100%)	
Hepatosplenomegaly	11 (78.6%)	6 (85.7%)	>0.99
SMA syndrome	2 (14.3%)	4 (57.1%)	0.52
Clinical findings			
History of AEDs medication	13 (61.9%)	13 (100%)	0.027 *
Gastrostomy	9 (42.9%)	9 (69.2%)	0.31
Tracheostomy	4 (19%)	2 (15.4%)	>0.99
Deafness	12 (57.1%)	5 (45.5%)	>0.99
Bed-ridden status	10 (47.6%)	11 (78.6%)	0.26

Abbreviations: TTE = transthoracic echocardiography; MR = mitral regurgitation; TR = tricuspid regurgitation; MV = mitral valve; US = ultrasonography; SMA = superior mesenteric artery syndrome; AEDs = antiepileptic drugs; * *p* < 0.05.

**Table 4 jpm-12-00665-t004:** Review of literature for age of death of patients with MPS III.

Age of Death (Years)	Reference	Study Disease	Number of Patients	Follow-Up Period
18.5	Lin et al. [27]	MPS IIIA, IIIB, IIIC	15	35 years
16.2	Malm et al. [26]	MPS IIIA	22	30 years
15.2	Meyer et al. [18]	MPS IIIA, IIIB	10	3 years
17 (60% probability)	Buhrman et al. [12]	MPS IIIA	46	13 years
15	Delgadillo et al. [17]	MPS IIIA	10	-

## Data Availability

All data generated or analysed during this study are included in this published article.

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
