# Peer review of "Natural History and Molecular Characteristics of Korean Patients with Mucopolysaccharidosis Type III"

_jpm, 2022, doi:10.3390/jpm12050665_

Round 1

Reviewer 1 Report

Kim and co-workers present us quite a detailed and nicely described natural history registry of the Korean patients with MPS III.

Overall, this is the first study describing the natural history of Sanfilippo in the Korean population. Furthermore, its molecular bases were also addressed, with the authors presenting a brief overview of the mutations detected in this population, on the different MPS III-related genes. The paper is well-designed and easy to follow and I do believe it offers relevant information, helping to raise awareness on MPS III and its major clinical symptoms. In general, natural history descriptions such as this one, are relevant contributions to the field, as they actively benefit drugs development. This is particularly relevant in the rare diseases field, where natural history information is usually not available or is incomplete for most diseases.

Therefore, I have no major doubts in recommending this manuscript for publication.

There are only a few minor issues that I would like to see corrected/addressed, which I list below:

  • On the Introduction section (page 2, line 15) – the reference that the authors quote (Héron et al., 2011) is not listed. Please include it.
  • Still on the introduction section (page 2, line 30) – while referring to the several disease-specific treatments for MPS III, which are either under development or have been addressed so far, the authors use the expression “have been studied”. Nevertheless, and considering that some of those approaches are still under development and studies are ongoing on a number of MPS III treatments, it would be more accurate to state “have been considered or are being studied/evaluated”.
  • On the Materials and Methods section (page 3, line 21), the authors describe the RP and SP phenotypes. For the SP phenotype they use the following sentence “the SP phenotype progresses more slowly”. I would add a remark such as “as one could anticipate by its designation” or “as the name implies”. This is just a personal opinion, feel free to follow it or not.
  • Still on the Materials and Methods section (page 3), sub-section 2.1 MP III subtype classification (line 1), I would recommend the authors to change “enzyme activity” to “enzyme activities”, as different enzymes were assessed, depending on the pathology.
  • Finally, and still on the Materials and Methods section (page 4), sub-section 2.3 Ethics Statement (line 2), I would recommend the authors to check whether they need to change their statement on informed consent from “patients and their parents” to “patients and/or their parents” (depending on the age at which the patients were enrolled in the study and interviews conducted).
  • On the Results section (page 4), sub-section 3.4 Genetic Analysis (pages 4, 5 and 6), I have a few comments on the figures the authors chose to present, and which I think could be significantly improved:
    • 1) For consistency reasons, and in order to help the readers get a general sense of the mutation analysis results, I would suggest the authors to use always the same colour scheme throughout the different figures. Meaning: chose a single colour to indicate frameshift mutations, and use it in every single image were you refer to that sort of variants. The same for nonsense, and so on. So that one may easily guess the bars refer to the same for of mutation in the different graphs (this is particularly relevant for figure 4);
    • 2) I would probably go for a single panel on the genetic analysis results, instead of separating the panel with the mutations location from that with the mutations’ type.
    • 3) This is a personal opinion, but I would probably chose a different graph type for the mutation profile and that with the mutation distribution troughout the different exons. My suggestion would be to keep the distribution by exons in bars, but present the mutation type profile as a pie chart.
    • 4) I am a bit unsure whether those “notable mutations” the authors refer to should be listed on figure 5. If they chose to do it, maybe they should be highlighted in a different way (grey, for example?).
  • Still on the Genetic Analysis section, I feel it would be nice to include a table with the whole genotypic data – meaning the genetic characterization of each patient (maybe on Supplementary Material). That sort of data is also of relevance for molecular geneticists, as helps us get a real overview of the genetic profiles within each population.
  • On the Discussion section (lines 126 and 127), the references on the other two studies on the Asian population are missing.
  • Still on that section, I am a bit unsure whether the first column on table 4 is necessary.
  • Finally, on the Conclusions section, I would suggest a small change on the order of the arguments presented. Instead of writing “This is the first study that analysed the genetic and clinical characteristics of MPS III patients in Korea. MPS IIIA is the most predominant in Korea. MPS III should be considered when a patient shows neurodegenerative symptoms such as language retardation, motor retardation, or hyperactivity”, I would have written “This is the first study that analysed the genetic and clinical characteristics of MPS III patients in Korea. MPS III should be considered when a patient shows neurodegenerative symptoms such as language retardation, motor retardation, or hyperactivity MPS IIIA is the most predominant in Korea.”

Throughout the manuscript, there are also a few typos and/or minor language issues, which must be corrected. Here are some examples:

  • On the abstract (page 1, line 14) – instead of “follow by”, the authors should have written “followed by”.
  • On the Materials and Methods section (page 3), sub-section 2.1 MP III subtype classification (line 8) – instead of “Sanger sequencing test was performed for gene mutations”, the authors should have written Sanger sequencing test was performed to search for gene mutations”, for example.
  • On the results section (page 4, line 101)

There are also some places where the authors pasted the name of the enzymes, for accuracy, but forgot to change the colour of the lettering. That should also be corrected:

  • On the Introduction section (page 2, lines 6,7 and 8);
  • On the Materials and Methods section (page 3), sub-section 2.1 MP III subtype classification (lines 2 and 6).
  • On the Results section (page 4), sub-section 3.4 Genetic Analysis, there are some issues that need correction. For example:
    • Line 99 – the “of” that precedes the enumeration of the 4 known MPS IIIB-causing mutations, should be deleted;
    • Line 101 – instead of “the siblings with IIIC”, the authors should have written “the siblings with MPS IIIC”
    • Lines 101 and 102 – the sentence on the molecular characterization of the MPS IIIC siblings should be edited. Suggestion: “The siblings with MPS IIIC were found to be compound heterozygous for two known HGSNAT mutations: (….)”.
    • Line 105 – the “C” that refers to cDNA position on the mutation nomenclature, cannot be written in capital letters.

Also, there seems to be an issue regarding the manuscript’s page numbering from page 6. Please check and correct accordingly.

Finally, I also have a general comment on the figures: their captions should be moved from top to bottom, according to the journal’s rules.

Reviewer 2 Report

Manuscript Number: jpm-1676517
Type of manuscript: Article

Title: Natural history and Molecular characteristics of Korean patients with Mucopolysaccharidosis type III

Journal of Personalized Medicine

Special Issue: Personalized Medicine in Rare Disease

The aim of this study was to investigate the natural history of MPS III in Korean patients and the clinical differences between IIIA and IIIB of the rapidly progressing phenotype and between survival and death groups. The authors reported that MPS IIIA is the most predominant in Korea. MPS III should be considered when a patient shows neurodegenerative symptoms such as language retardation, motor retardation, or hyperactivity. There were no differences of clinical characteristics between patients with MPS IIIA and IIIB except for age at first walking. Early diagnosis is important to increase the survival rate, and new therapeutic strategies will significantly change the course of the disease. Their findings can be used to develop quality of care strategies and provide guidance for clinical trial endpoint evaluations.

This article is very important and well-written. Since to this date, it is the first study that analyzed the genetic and clinical characteristics of MPS III patients in Korea, this paper is worth publishing. There are only some minor points needed to be clarified or revised.

  • Page 6, 1. Clinical Presentation: The whole paragraph was shifted to the right side of this page. Please check.
  • Figure 5: “Frame Shift” in Figure 5A versus “Frameshift” in Figure 5B. Please check and uniform the style of this term.
  • Page 7, Table 4. Review of literature for age of death of patients with MPS III. The authors are suggested to add the data of the following fine publication which follow-up for 35 years (1985-2019) into this Table. (Lin HY, Lee CL, Chang CY, Chiu PC, Chien YH, Niu DM, Tsai FJ, Hwu WL, Lin SJ, Lin JL, Chao MC, Chang TM, Tsai WH, Wang TJ, Chuang CK, Lin SP. Survival and diagnostic age of 175 Taiwanese patients with mucopolysaccharidoses (1985-2019). Orphanet J Rare Dis. 2020 Nov 7;15(1):314. Table 1. The mean and median age of diagnosis and at death for patients with different types of MPS).
  • Page 8, Supplementary Materials: Figure S2 (A, B and C): The authors described that “There are age of speech loss, age of walking ability loss, and age of undergone gastrostomy in MPS IIIA and IIIB is shown, respectively. However, “Survival probability” was shown on all these three figures. Please change the descriptions shown on the Y-axis of these three figures.
